# SANTAVAC^TM^: Summary of Research and Development

**DOI:** 10.3390/vaccines7040186

**Published:** 2019-11-17

**Authors:** Petr G. Lokhov, Mikayel Mkrtichyan, Grigor Mamikonyan, Elena E. Balashova

**Affiliations:** 1BioBohemia Inc, 548 Market Street #45666, San Francisco, CA 94104, USA; 2Institute of Biomedical Chemistry, Pogodinskaya st., 10/8, Moscow 119121, Russia; balashlen@mail.ru; 3The Loop Immuno-Oncology Laboratory, Lombardi Comprehensive Cancer Center, Georgetown University Medical Center, 4000 Reservoir Road, NW, Suite 120, Washington, DC 20057, USA; mm4530@georgetown.edu; 4Clinartis LLC, 1909 Tyler Street, Suite 601, Los Angeles, FL 33020, USA; gmamikonyan@clinartis.com

**Keywords:** SANTAVAC, cancer vaccine, proteomic footprint, antiangiogenic vaccine, dendritic cell vaccine, peptide vaccine, autologous cellular therapy

## Abstract

SANTAVAC is an antigen composition developed via proteomics and cell culture technology that is intended for the development of cancer vaccines against various solid tumors. Its mechanism of action is based on the heterogeneity of endothelial cells, the polypeptides of which are similar to the surface antigens of tumor-vessel cells, allowing targeted destruction by vaccination. While research and development work with SANTAVAC is ongoing, the existing data provide strong evidence that allogeneic SANTAVAC is an ideal candidate for the development of cancer vaccines with significant efficacy and safety. The SANTAVAC compositions described here demonstrated the ability to inhibit the growth of tumor vessel-specific endothelial cells up to 60 fold, with minimal effect on normal vasculature. Innovation, background, description of product development, and summary of nonclinical studies with SANTAVAC to date are presented in this review.

## 1. Introduction

Cancer immunotherapy is a long-standing and rapidly developing area in which new methods of cancer treatment are being developed through increased understanding and use of the immune system [1]. Today’s cancer immunotherapy offers cancer vaccines, one of the field’s most exciting developments over the last decade. The first anti-cancer vaccines comprised whole-cell preparations used in combination with an adjuvant to enhance the immune response. Later studies revealed a wide range of potential targets for cancer vaccines, among them well-known tumor-specific antigens such as mutated oncoproteins (p53, ras, and B-Raf), overexpressed antigens (mucin 1, human epidermal growth factor receptor, carcinoembryonic antigen, and a-fetoprotein), tissue line and differentiation antigens (prostatic acid phosphatase, prostate-specific antigen, glycoprotein 100, and MART-1), and others [2,3,4]. Vascular endothelial growth factor receptor-directed vaccines are known to target the tumor microenvironment [5]. Despite all of this progress, only three cancer vaccines have been approved by the FDA. TheraCys (Sanofi Pasteur) was approved in 1990 for the treatment and prophylaxis of urothelial carcinoma [6]. PROVENGE (Dendreon Corporation) was approved in 2010 and represents autologous cellular immunotherapy indicated for the treatment of metastatic prostate cancer [7]. T-VEC (Amgen) is a genetically modified oncolytic viral therapy approved in 2015 for the treatment of advanced melanoma [8]. Other cancer vaccine trials have been negative, despite these therapies being generally well tolerated with favorable side-effect profiles.

Modern cancer vaccines belong to several main groups: Whole tumor cells, peptides and proteins, recombinant vectors, dendritic cells, gangliosides, and genes. Each group has its own advantages and disadvantages. As of November 2019, there have been 329 active clinical trials of cancer vaccines (www.clinicaltrials.gov). Ongoing trials aim to increase the effectiveness of cancer vaccines by targeting new tumor antigens or using vaccines in combination with other therapeutic approaches. Many preclinical and clinical studies have shown that cancer vaccines can activate effector T cells against tumor antigens and kill different types of tumor cells with minimal toxicity.

The most accurate targeting of the immune system by vaccination is provided by native antigens, which are almost identical to the target cell antigens. The efficacy of native antigens for anti-cancer vaccination has been studied for almost 30 years, involving scientific and preclinical studies as well as clinical trials in humans [9,10,11]. The most common way to use native antigens in immunotherapy is whole-cell vaccination. Besides the expected accurate targeting, benefits of such vaccinations over those that target separate native or synthesized antigens are that multiple antigens, including ones that are yet to be discovered, can be targeted by the immune system. That is especially important in anticancer vaccination, where cancer escape is a big challenge [12]. Unfortunately, whole-cell vaccination did not produce any significant therapeutic benefits [13,14]. One possible reason for this is that whole cells pose to the immune system a plethora of nonspecific antigens, such as house-keeping proteins, nucleic acids, carbohydrates, and lipids, that are expressed in all cells. As a result, whole-cell preparations contain only trace amounts of the target antigens buried in an abundance of noise [15], insufficient for eliminating cancer cells. Therefore, whole-cell anti-cancer vaccination is a thing of the past, and further developments are needed to release the untapped potential of native cellular antigens in cancer immunotherapy.

Further development of cell-based preparations may be directed toward the elimination of intracellular antigens common to all cells. The remaining surface antigens, mainly proteins and carbohydrates, would be natural targets for immunotherapy due to their permanent availability for humoral and cellular immune responses. The role of cell surface antigens in targeted immune responses has been well investigated and reported [16,17,18,19,20,21]. Briefly, cancer cells escape immune surveillance by maintaining a cell surface that lacks immunodominant antigens [21,22] and presents immunosuppressive ones [23,24,25,26,27]. However, the surfaces of cancer cells still show some evidence of antigenicity. For example, antibodies produced by an organism after whole-cell vaccination were found to selectively bind with surface targets of the injected cells [28,29,30]. In other studies, the removal of cell surface molecules with protease resulted in the reduction or elimination of cell antigenicity [31,32,33,34,35]. 

Together, these data demonstrate that the cell surface defines the antigenic properties of live cells. Therefore, the next step for discovering efficient cell-based vaccines is to separate cell surface targets, i.e., the ‘antigenic essence’, from other cellular contents, and use them in the development of cancer vaccines.

## 2. Innovation 

The key idea is to use a protease to collect a set of target cell surface antigens cleared from undesired intracellular content. It is expected that if target surface antigens are accessible to the immune system, then they are also accessible to the protease. Although the isolation of cell surface antigens for vaccination, and the use of protease for antigen separation, are not novel ideas [36,37,38], the preparation of a true antigenic essence became possible only with advances in proteomics. As mentioned above, the antigenic essence is present only in trace amounts in whole-cell preparations. Treatment of live cells with a very low concentration of high-purified (proteomics grade, protected from autolysis) trypsin avoids damage to cell wall permeability and collects antigens only from the surface. The resulting antigens are neither contaminated by intracellular content nor overridden by trypsin, its admixtures, and products of autolysis [15]. In order to automate the composition of antigenic essence, antigens are analyzed by an advanced proteomics approach, based on peptide mass spectrometry, that yields a highly specific antigen composition code (Appendix A). This code defines cancer cell death from cytotoxic lymphocytes with mathematical precision in an in vitro model of human vaccination. 

The antigenic essence was named SANTAVAC (a Set of All Natural Target Antigens for Vaccination Against Cancer) and used as an improved derivative of whole-cell preparations. SANTAVAC formulations consist only of target antigens and have specific code-related compositions (hereinafter referred to as the SANTAVAC code), allowing for precise in vitro development of a family of new cancer vaccines that can replace the obsolete whole-cell vaccines.

## 3. SANTAVAC Equivalence to Cell Antigens 

To confirm that SANTAVAC is sufficient for developing effective vaccines, the immunogenic properties of SANTAVAC were compared to those of whole cancer cells. In cytotoxicity assays, human cytotoxic T lymphocytes (CTLs) were incubated with target adenocarcinoma cells (MCF-7) and stimulated with SANTAVAC or whole cancer cell lysate. It was observed that SANTAVAC was 10–40% more effective at inducing cytotoxicity than the whole cell lysate (Figure 1), even though the total protein concentration was substantially lower (whole cell lysate: 270 µg/mL vs. SANTAVAC: 2 µg/mL) [39]. The higher levels of cytotoxicity associated with a significantly lower protein concentration demonstrate that SANTAVAC composition is free from intracellular contaminants, and includes a comprehensive set of antigens responsible for targeting immune response toward cells. It was concluded that these results provided a proof-of-concept that the proteolytic treatment of live cancer cells can release a complete set of antigenic targets, i.e., SANTAVAC, that induces an anti-cancer immune response that exceeds that of whole cancer cells.

To support further development of antiangiogenic SANTAVAC vaccines, the immunogenic properties of SANTAVAC were compared to those of whole endothelial cells (ECs). Figure 2 represents a schema of antiangiogenic SANTAVAC production for peptide or dendritic cell-based vaccines.

In cytotoxicity assays, EC lysate and SANTAVAC had total protein concentrations of 135 and 2 µg/mL, respectively [41]. Despite this difference in concentration, SANTAVAC was ~20% more effective at stimulating immune cells. Moreover, SANTAVAC obtained from tumor-activated ECs was able to stimulate an immune response specifically toward tumor-activated ECs. Based on these results, it was concluded that SANTAVAC provides a comprehensive set of surface antigens that are able to induce targeted, immune-mediated cytotoxic effects against tumor ECs. These findings represent a successful strategy to produce safe and pure antigens for the design and production of SANTAVAC-based antiangiogenic vaccines.

## 4. Method to Control SANTAVAC Composition 

SANTAVAC is a product of cell culture technology. Cultivated cell studies have indicated that cross-contamination between cell lines is widely prevalent and continues to be a major problem [42,43,44,45]. Moreover, cultivated mammalian cells have a finite mitotic lifespan [46,47], which is followed by cellular degeneration and modification of their surface molecular profile [48,49]. Accordingly, cultivated cells intended for therapy, or for the isolation of specific compositions of cellular antigens, as is the case for SANTAVAC, must be authenticated, and cellular products should be qualified in order to exclude undesired changes in their composition. 

Authentication and characterization of primary cultures are mainly performed by the cell culture vendors. Analysis of SANTAVAC composition is more complex. SANTAVAC is a novel composition, which contains a very low overall concentration of different peptides (Table 1), which is considered a trace amount. Therefore, a special approach is required to analyze and control SANTAVAC composition. Proteolytically-separated cell surface antigens (i.e., SANTAVAC) can be analyzed using mass spectrometry. This technique is related to proteomic footprinting and represents a simple approach for the authentication and characterization of cells on the subtype level (see Appendix A, Figure A1) [50]. By comparing the composition of SANTAVAC code with a proteomic footprint of the reference cells, the SANTAVAC composition and origin can be easily authenticated (Figure 3). Moreover, any divergence in this comparison can reveal changes in antigen composition that may have occurred. As an example, the control of antiangiogenic SANTAVAC composition is presented in Section 8.

## 5. SANTAVAC Composition and Strength of Immune Response

To reveal the connection between SANTAVAC composition and immune response as measured by cytotoxicity assays (CTA), cell footprints of target MCF-7 cells and SANTAVAC codes were correlated with CTA results, wherein cell surface profiles of target MCF-7 cells were gradually changed by drug selective pressure (Figure 4). Results clearly showed that the rate of cell escape from immune response depends on the similarity between SANTAVAC composition and footprints of target cells. This relationship between escape rate and footprint similarity was linearly approximated, with *R*^2^ almost equal to 1 [51]. 

The high degree of correlation between CTA effectiveness and SANTAVAC/target similarity (i) confirms that SANTAVAC represents the antigenic essence of cells with a complete set of target antigens, (ii) supports the multivariable nature of SANTAVAC code and target cell footprint, which count hundreds of variables (peptides), and (iii) provides a basis for future use of precise mathematical models in SANTAVAC vaccines development (the effect of fluctuations in the concentration of individual antigens is avoided). 

These results also demonstrate that the set of antigens expressed on the surface of cancer cells can be substantially modified under the selective pressure of chemotherapeutic drugs (several drugs were used in this study [51]). Such adaptive changes often allow cancer cells to escape the immune response. One strong point in favor of SANTAVAC is that its composition may be developed to take such changes into account, and thus target even those evasive cancers. 

## 6. Antiangiogenic SANTAVAC Vaccines

### 6.1. Background, Rationale, and Significance 

Studies to inhibit angiogenesis for therapeutic treatment mainly related to the oncological field, and resulted in the FDA approval of several inhibitors. The first one to be approved was bevacizumab (Avastin), which has shown a clinical benefit for several cancers [52]. Later were approved aflibercept and the small-molecule tyrosine kinase inhibitors sunitinib, sorafenib, and pazopanib. These and other approved anti-angiogenic drugs demonstrate only limited clinical benefit [53]. Clinical studies have shown that survival without progression can be achieved, but in most cases, this does not affect overall survival. Moreover, the early stages of cancer do not appear to respond to such treatment [54].

Vaccination against the tumor vascular system may overcome the limitations inherent in modern antiangiogenic drugs because it combines the advantages of both immunotherapy and antiangiogenesis. Currently, there are a number of anti-angiogenic vaccination targets, including soluble pro-angiogenic factors (most studies have focused on VEGF), endothelial matrix, EC membrane, tumor endothelial markers (TEMs), FGFR-1, VEGFR1, VEGFR2, and endoglin [53]. Clinical trials are currently testing the most promising of these anti-cancer vaccination approaches, but with limited success [55]. It has been hypothesized that the limited success of anti-angiogenic vaccination is due in most cases to targeting tumor-derived growth factors and their receptors [55,56,57]. This approach gives growth advantages to mutated tumor cells which have the capacity to use alternative growth factor pathways to induce angiogenesis. It has been suggested that the direct targeting of tumor ECs, given their genetic stability, should surpass most existing drugs [53]. Therefore, the identification of tumor-associated EC markers [58,59,60] and multitargeting are currently of key importance and may overcome intrinsic limitations [55]. In light of this, reassessment and improvement of whole EC-based vaccines seem reasonable.

The first work related to the use of whole EC preparations as vaccines was published by Wei et al. [61]. They showed in an animal model that immunization with xenogeneic ECs inhibited tumor growth both prophylactically and therapeutically. Immunization with allogeneic cells had no such effect. Okaji et al. [62,63] demonstrated in an animal model that vaccination with autologous ECs induced an immune response against endothelium and inhibited the formation of lung metastases. Notably, the autologous vaccine was shown to be superior in terms of immunogenicity and inhibition of metastasis when compared with a similarly constructed xenogeneic vaccine. Importantly, the vaccine was able to induce antibody and cellular responses without adverse events. However, no clear correlation could be found between the immunological and anti-tumor responses. Taking into account the promising results of this study, correction of the main deficiency of whole-cell vaccines (an excess of undesirable antigens, which leads to insufficient specificity and diluting of the immune response) is critical to the development of anti-angiogenic cancer vaccines.

### 6.2. Development of Antiangiogenic SANTAVAC Vaccines

Endothelial cells (ECs) line the inner surface of blood vessels and constitute a selective barrier between blood and tissue. They play a critical role in a variety of physiological and pathological processes [64,65]. The importance of ECs in the context of cancer has been extensively investigated [66]. In 1945 it was reported that, in an animal model, an engrafted tumor recruits capillaries from the host to support its feeding and growth [67]. This finding gave rise to an entire field of research aiming to inhibit new blood vessel formation [66]. Among these studies was vaccine development based on the discovery that endothelial antigens were overexpressed in tumors relative to peripheral tissues. Notably, vaccination against EC antigens in tumors offers the additional benefit that ECs are genetically more stable, and therefore less likely to develop escape mutations, than cancer cells [68]. Considering that cancer cells appear able to escape the SANTAVAC-induced immune response (Figure 4), the development of antiangiogenic SANTAVAC is an urgent priority. Moreover, the ratio of ECs to cancer cells in tumors is approximately 1:100. Destruction of a small number of ECs can lead to vascular obstruction and arrested tumor growth, because vascular integrity is essential to tumor feeding [69,70,71,72].

The molecular phenotype of ECs in the microvasculature is tissue type-specific [73,74], and if the microvasculature involved is feeding a tumor, the EC phenotype also becomes tumor cell type-specific [75,76,77]. This heterogeneity in phenotype provides grounds for designing antiangiogenic SANTAVAC. The influence of different tumor cells on the EC surface profile was investigated by culturing human microvascular endothelial cells (HMECs) with tumor-conditioned medium containing all stimuli released by cancer cells that may affect the EC phenotype. 

The use of HMECs in studies was justified by data showing that tumor vascularization primarily involves microvasculature. When compared to ECs of large vessels, HMECs exhibit functional and phenotype differences, including in response to stimulation [78,79,80,81,82]. The use of HMEC primary cultures allowed for the investigation of the natural surface phenotype and response to stimuli from cancer cells. HMECs for these experiments were derived from easily accessible subcutaneous fat. To exclude potential contamination by other cell types, including fibroblasts and mesothelial cells [83,84,85], the CD31+ HMECs were isolated using magnetic beads [86]. Accurate isolation was then confirmed by immunofluorescence staining for CD31 and FACS analysis [85,87]. The purity of established HMEC cultures was considered high (~90%) [88] and corresponded to that of commercially available high-grade primary cultures [89]. 

The effect of similarity between SANTAVAC code and footprint of target HMECs on their escape from the immune response was determined with cytotoxicity assays, as an in vitro model of vaccination. To this end, human CTLs were incubated with SANTAVAC-loaded DCs. It was observed that autologous SANTAVAC stimulated the killing of target cells more effectively than allogeneic antigens (Figure 5) [90,91].

Further, the efficacy of immune system targeting by autologous SANTAVAC with different codes was examined. Plotting SANTAVAC/target similarity against the results of the cytotoxicity assay revealed a strong linear relationship, with *R*^2^ values equal or almost equal to 1, following the equation [90]:(1)n = k∗r + bwhere *n* is the number of total viable target cells in cytotoxicity assays (representing target cell escape), *r* is the correlation between profiles of target cells and cells used to develop SANTAVAC, *b* represents contributions to the immune response independent of the profiles of target cells and cells used to develop SANTAVAC, *k* defines the intensity of the immune response.

It was suggested that *k* reflects the degree of tumor-induced changes in HMEC surface and *b* reflects the immunogenicity as a result of these changes. Moreover, it was revealed that *k* and *b* were dependent on each other according to the linear equation [90]:(2)$$b\;=\;-0.67*k\;+\;9754\;(R^{2}\;\mathrm{of}\;\mathrm{linear}\;\mathrm{approximation}\;\mathrm{is}\;0.99)$$

Thus, it was found that the immunogenicity of target cells is inversely proportional to the intensity of tumor-induced changes to the cell surface. Notably, equations are valid for all antiangiogenic SANTAVAC compositions. That is, variation in cancer cells causes variation in the strength, not the nature, of changes to the HMEC phenotype. That critical finding provided a strong foundation for designing a universal SANTAVAC composition that can be highly specific and still able to affect different tumors. Therefore it is possible to develop allogeneic antiangiogenic SANTAVAC which excludes the patient’s own biomaterial from vaccine preparation, thereby simplifying production and facilitating clinical implementation.

Previously, a high killing rate was observed for allogeneic SANTAVAC produced from HepG2-stimulated HMECs, which targets HMECs stimulated by human breast adenocarcinoma cells (Figure 5a). The observed efficacy provided a therapeutic window in which HMECs of tumors could be killed before those of normal tissues are affected. To determine the optimal stimulation required from HepG2 cells, the HepG2-conditioned medium was added to HMECs at different concentrations [92]. After three days in culture, cells were counted to determine the concentration of tumor-conditioned medium that induced weak (stimulated cells slightly higher than unstimulated controls), moderate (half of the maximum), and strong (maximum) stimulation of HMECs (Figure 6) and production of SANTAVAC. 

Cytotoxicity assays were used to calculate in vitro efficacy of the allogeneic SANTAVAC compositions, namely, efficacy type I (denoted as “efficacy I”), calculated as a ratio of the number of non-stimulated cells in control (i.e., HMEC^0%^) to the number of tumor-stimulated cells in the experiment. Efficacy type II (denoted as “efficacy II”) was calculated as a ratio of the number of tumor-stimulated cells in control (i.e., HMEC^5%^, HMEC^15%^, or HMEC^25%^) to tumor-stimulated cells in the experiment. SANTAVAC^5%^, SANTAVAC^15%^, and SANTAVAC^25%^ compositions produced from HMECs stimulated with 5%, 15%, or 25% tumor-conditioned medium (i.e., from HMEC^5%^, HMEC^15%^ and HMEC^25%^, respectively) were tested (Figure 7).

Subtle improvement in cytotoxicity was observed when CTLs were stimulated with control DCs (loaded with surface antigens collected from fibroblast primary culture). SANTAVAC^15%^-loaded and SANTAVAC^25%^-loaded DCs elicited effective immune responses as measured by high death rates of target HMECs (Figure 7). Notably, CTLs stimulated with SANTAVAC^15%^ were most effective against HMEC^15%^ target cells (almost all target cells were dead). The target HMECs^25%^ were most efficiently killed by CTLs stimulated with SANTAVAC^25%^. SANTAVAC^25%^ efficacy was very low against weakly stimulated target HMECs^5%^. SANTAVAC efficacy I indicates that in vitro modeled vaccine safety was 17.3, achieved using SANTAVAC^15%^, and target HMEC^15%^. SANTAVAC efficacy II indicates that the in vitro modeled capacity to arrest tumor growth was ∼60, also achieved by SANTAVAC^15%^, target HMEC^15%^. 

Obtained efficacy data suggest SANTAVAC^15%^ as a candidate for therapeutic cancer vaccines. Since SANTAVAC^25%^ exhibited less strength of action than SANTAVAC^15%^, it can be considered as a candidate for preventive cancer vaccines. It is expected that SANTAVAC^25%^ will not affect normal tissues or tissues weakly stimulated by tumor vasculature, resulting in very low side effects. 

While more mechanistic and functional studies using SANTAVAC are ongoing, the in vitro data presented above demonstrate that allogeneic SANTAVAC is an ideal candidate for the development of therapeutic and preventive vaccines with outstanding efficacy and safety. The SANTAVAC formulation described inhibited tumor vessel-specific EC growth up to 60-fold. Moreover, it had minimal effects on normal vasculature.

## 7. Antiangiogenic SANTAVAC Production and Physical Properties

The primary culture of HMECs (from derma or fat tissue) is used for the production of SANTAVAC (Figure 2 and Appendix B). First SANTAVAC composition intended for trials is a biological entity for ex vivo loading of DCs for autologous cellular immunotherapy of cancer. This SANTAVAC product is a sterile, transparent balanced salt solution (buffer) with low salt levels and very low (glyco)peptide levels (trace quantity), with a composition that is highly specific for cell type and cell molecular subphenotype. It has a ready-to-use formulation for loading of DCs. Its peptide composition, which is related to the particular cells from which the peptides originated, defines its immune response targeting activity directly. The main characteristics of the SANTAVAC product intended for loading of DCs are shown in Table 1. 

## 8. Control of Antiangiogenic SANTAVAC Composition 

Antiangiogenic SANTAVAC composition is controlled by peptide mass spectrometry according to proteomic footprinting technology, as described previously [90]. SANTAVAC solution was desalted using ZipTip_C18_ (Millipore Corp.), and MALDI samples were prepared using a standard “dried droplet” method with 2,5-dihydroxybenzoic acid as the matrix. All mass spectra were acquired on a MALDI-TOF mass spectrometer in linear positive ion mode. The mass spectrometer was set up for priority detection of ions with the m/z range from 600 to 3500 at a mass accuracy of 80–100 ppm. All peaks above noise level were selected to generate peak lists that represent the cell surface profiles (known as cell proteomic footprints or SANTAVAC code [50]). 

Resultant lists of peak intensities were compared with the codes of any other two non-endothelial cells (e.g., MCF-7 and LnCap cells) and six HMECs (two of which were activated by HepG2 cells, two by any other cancer cells, and two from non-activated HMECs). Two peaks were considered to be related to the same ion if their mass difference does not exceed 0.2 Da. Pooled intensities were processed by PCA. Projections of peak intensities on the first three principal components are used to visualize the divergence among cell surface profiles. The distances between profiles on the PCA plot are also depicted as a dendrogram. The tested code should fall in the same cluster with code obtained from HMECs activated by HepG2 (Figure 8) [50]. This confirms that SANTAVAC was produced as required, is cell type-specific, and its composition is qualified for vaccine development.

## 9. Development Plans

BioBohemia, as a main sponsor of research and development (R&D), is expanding nonclinical (Appendix C, Table A1) and clinical development to the United States (including production and quality control testing) with the objective of obtaining market authorization in the United States. BioBohemia’s strategy is based on development and rapid clinical entry of a cancer vaccine based on SANTAVAC, demonstration of clinical safety and efficacy of this approach, and follow-on development of additional immunotherapeutic candidates. This strategy will allow for the leveraging of initial development efforts of the most advanced candidate to allow expedient transition for the other candidates. BioBohemia’s therapeutic pipeline includes technologically distinct candidate products, each formulation type having its advantages and disadvantages:*Therapeutic SANTAVAC^15%^-loaded allogeneic dendritic cell vaccine* is an autologous cell therapy for which primary culture HMECs can be purchased to produce SANTAVAC. DCs should be obtained from patient blood.*Preventive SANTAVAC^25%^-loaded allogeneic dendritic cell vaccine*, similar to therapeutic vaccine, is an autologous cell therapy for which commercially available HMECs and DCs from patient blood can be used.*Therapeutic SANTAVAC^15%^-loaded autologous dendritic cell vaccine* is an autologous cell therapy prepared from the patient’s own biomaterial. HMECs for primary culture can be isolated from abdominal adipose by needle biopsy. DCs are obtained from the blood of the patient.*Therapeutic SANTAVAC vaccine* is allogeneic SANTAVAC^15%^ mixed with adjuvant(s).*Preventive SANTAVAC vaccine* is allogeneic SANTAVAC^25%^ mixed with adjuvant(s).

In addition to encompassing technologically distinct candidates, the therapeutic pipeline encompasses separate formulations that can also target different types of cancer. Notwithstanding, the development programs are all highly focused on one target: an antigen composition developed by SANTAVAC technology and intended for the development of cancer vaccines against solid tumors. This circumstance simplifies the overall R&D efforts and creates synergy among all of the programs, allowing the programs and their associated achievements to leverage each other. 

BioBohemia is now extending the nonclinical development of SANTAVAC, the next step will consist of testing the final SANTAVAC formulations in animal models. Although the animal model is disclosed in patents to protect all SANTAVAC formulations in general, the final formulations have not yet been tested in animals according to preclinical trial requirements. Antiangiogenic SANTAVAC is a highly specific composition of human allogeneic antigens, and its therapeutic properties are best demonstrated upon allovaccination of humans or appropriate animal models (i.e., tumor xenografts vascularized with human microvascular endothelial cells). Previously, human blood vessels were engineered in immunodeficient mice using human endothelial cells. This method allows for a quantitative investigation of anti-cancer drug effects on human angiogenesis in a murine host [93,94] and provides scientific justification for testing SANTAVAC in a mouse model. Mice with tumors that have been vascularized with human vessels exhibit the required key characteristics of the human disease, and any effects of SANTAVAC are expected to be predictive of its effects in humans. According to FDA requirements (‘Animal Rule’ guidelines), the efficacy of SANTAVAC will be demonstrated in two animal (mouse) models. It is expected that after animal testing and other common tests describing SANTAVAC composition, toxicity, stability, etc., two peptide SANTAVAC vaccines will be registered by FDA as innovative new drugs, and two others as biological entities for autologous cellular therapy. This will allow further use of SANTAVAC vaccines in human trials.

## 10. Conclusions

Multitarget vaccination against tumor vessels should outperform most FDA-approved antiangiogenic drugs. In light of this, reevaluation and improvement of whole EC-based vaccines seems reasonable. Antiangiogenic SANTAVAC is the antigenic essence of ECs, which preserves all the benefits of whole cells without their shortcomings, and demonstrates very promising efficacy in model experiments. Currently, SANTAVAC R&D is finished, and the vaccine has moved to preclinical stages in the United States.

## 11. Patents

Lokhov P.G. “Method for producing an antitumoral vaccine based on surface endothelial cell antigens”, 2007, Eurasian patent №009327.Lokhov P.G. Balashova E.E. “Method for testing cell culture quality”, 2007, Eurasian patent №009326.Lokhov P.G. Balashova E.E. “Method for producing an antitumoral vaccine”, 2009, Eurasian patent №011421.Lokhov P.G. “Method for producing an antitumoral vaccine based on surface endothelial cell antigens”, 2012, Japanese patent №5154641.Lokhov P.G. “Method for producing an antitumoral vaccine based on surface endothelial cell antigens”, 2013, Korean patent №10-1290641.Lokhov P.G. “Method for producing an antitumoral vaccine based on surface endothelial cell antigens”, 2015, European patent №2140873 (Protection in Switzerland/Liechtenstein, Germany, Spain, France, the United Kingdom, Ireland and Italy).Lokhov P.G. “Method for producing an antitumoral vaccine based on surface endothelial cell antigens”, 2017, USA patent №9844586.

### Patents, Related to Same Technology Based on Cancer Cells

8.Lokhov P.G. “Antitumoral vaccine, method for producing an antitumoral vaccine and method for carry out antitumoral immunotherapy”, 2007, Eurasian patent №009326.9.Lokhov PG. “Tumor vaccine, a method for producing a tumor vaccine and a method for carrying out antitumor immunotherapy”, 2013, Japanese patent №5172864.10.Lokhov P.G. “Tumor vaccine, a method for producing a tumor vaccine and a method for carrying out antitumor immunotherapy”, 2012, Chinese patent №CN101636174 B.

The trademark SANTAVAC™ was registered in International Register of Marks maintained under the Madrid Agreement and Protocol. Registration number 1326157. Date of registration 12 September 2016.

## Figures and Tables

**Figure 1 vaccines-07-00186-f001:**
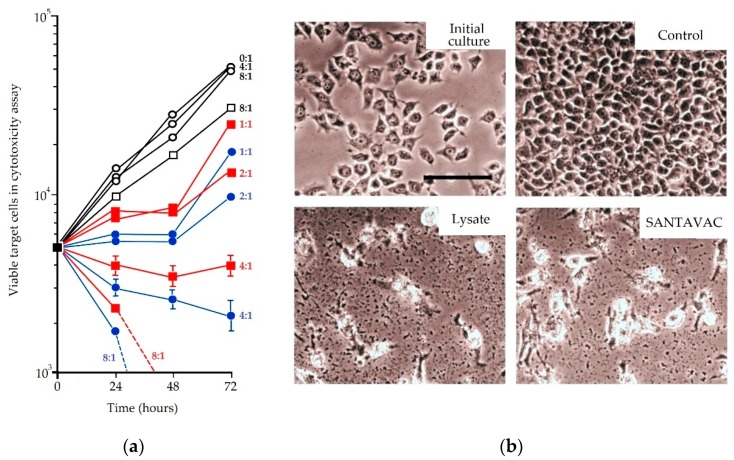
Antigenic equivalence of Set of All Natural Target Antigens for Vaccination Against Cancer (SANTAVAC) to whole cell lysate. (**a**) Cytotoxicity of effector CTLs against MCF-7 cells. Target MCF-7 cells were seeded in tissue culture plates, and then effector CTLs were added at CTL:MCF-7 ratios ranging from 1:1 to 8:1. A ratio of 0:1 indicates growth of MCF-7 cells to which effector CTLs were not added. (●) MCF-7 cells incubated with CTL that had been stimulated with SANTAVAC-loaded DCs. (■) MCF-7 cells incubated with CTL that had been stimulated with lysate-loaded DCs. (□) MCF-7 cells incubated with CTL that had been stimulated with unloaded DCs. (○) MCF-7 cells grown either alone, or with unstimulated PBMC. Points represent the mean value of three identical measurements. As an example, the standard deviation for the 4:1 ratio is shown (similar values for standard deviation were obtained using other CTL:MCF-7 ratios). (**b**) Images for target MCF-7 cells: ‘Initial culture’: Initial cell culture of MCF-7 cells. ‘Control’: Control MCF-7 cells grown alone, ‘Lysate’: MCF-7 cells incubated with CTLs that had been stimulated with dendritic cells loaded with whole cell lysate, ‘SANTAVAC’: MCF-7 incubated with CTLs that had been stimulated with dendritic cells loaded with SANTAVAC. MCF-7 cells were incubated with effector CTLs at a ratio of 4:1. On the third day, target cells were carefully washed and imaged using an inverted phase-contrast microscope (scale bar: 50 µm). SANTAVAC induces 10–40% more cytotoxic activity in CTLs than whole cell lysate, even though the total protein concentration of SANTAVAC formulation was substantially lower (whole cell lysate: 270 µg/mL vs. SANTAVAC: 2 µg/mL). Adapted from [39].

**Figure 2 vaccines-07-00186-f002:**
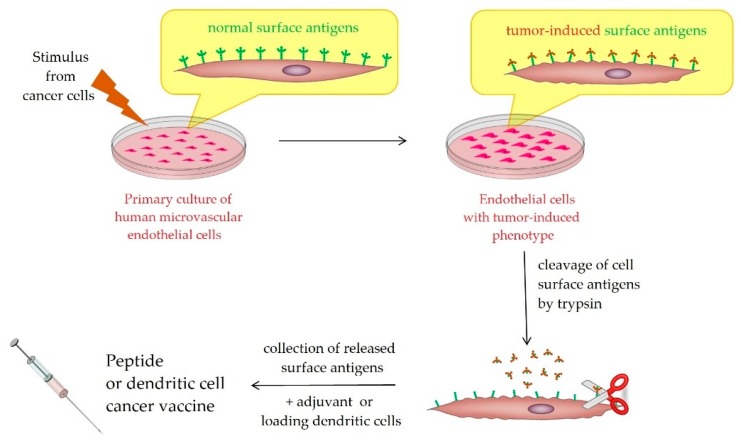
Schematic representation of antiangiogenic SANTAVAC production for peptide or dendritic cell-based vaccine. Adapted from [40].

**Figure 3 vaccines-07-00186-f003:**
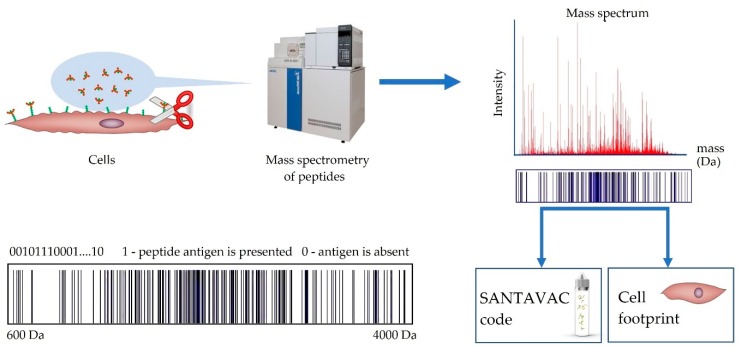
Workflow of cell proteomic footprinting. Adherent cell culture, after washing away traces of culture medium, is treated with a protease. Released fragments of the cell surface proteins are collected and analyzed by mass spectrometry. The set of obtained peptide molecular weights represents the SANTAVAC code and cell culture proteomic footprint. Comparison of these codes allows for authentication of the cells used to generate SANTAVAC, and also reveals any changes in its composition. This proteomic footprinting method was developed as a part of research and development (R&D) for SANTAVAC. Adapted from [50].

**Figure 4 vaccines-07-00186-f004:**
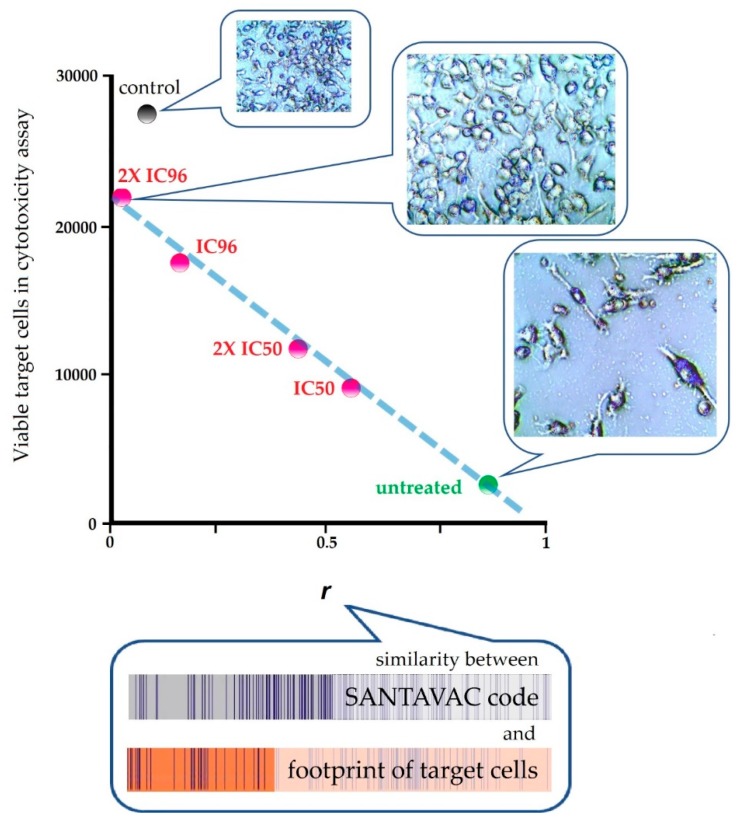
Escape of cancer cells from the SANTAVAC-mediated immune response in CTA as a result of cell surface profile changes induced by the selective pressure of drug treatment. Points are presented for MCF-7 target cells that were untreated or treated with either a single dose of IC96 or IC50 etoposide or two separate doses (‘2X’) of IC96 or IC50 etoposide. Linear approximations for points related to MCF-7 cells are shown. Tested SANTAVAC consists of surface antigens of untreated MCF-7 cells. The average number of viable cells in three wells is presented. Correlation coefficients (*r*) were calculated for SANTAVAC code and footprints of target cells (i.e., SANTAVAC/target similarity). ‘Control’ corresponds to SANTAVAC produced from HepG2 cells. Adapted from [51].

**Figure 5 vaccines-07-00186-f005:**
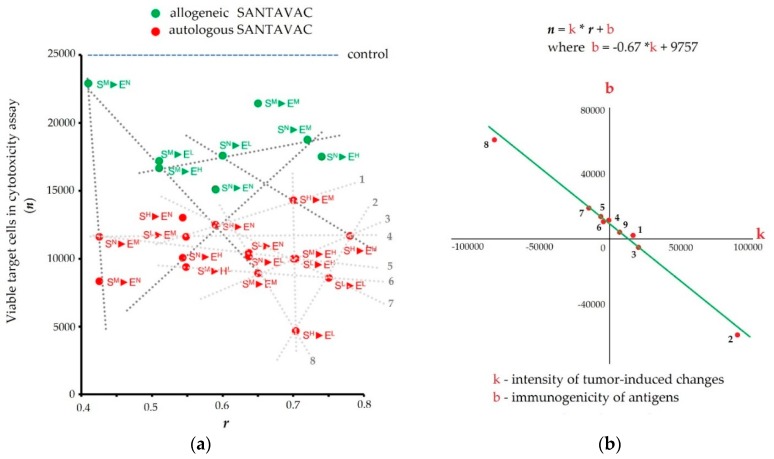
Cytotoxicity of effector CTLs against target human microvascular endothelial cells (HMECs). (**a**) Cytotoxicity plotted vs. correlation of SANTAVAC composition (code) and cell surface profile (footprint) of target cells. The average number of viable target cells in three wells is presented. Correlation coefficients (*r*) were calculated for SANTAVAC code and footprint of target HMECs (i.e., SANTAVAC/target similarity). ‘1^x^►2^x^’: First letter corresponds to SANTAVAC and HMECs used as a source of antigens to produce this SANTAVAC (superscript), and the second letter corresponds to the target HMECs used in the cytotoxicity assay. Red and brown letters correspond to autologous and allogeneic antigens, respectively. Letters are used to identify HMECs stimulated to grow in the presence of an EC growth supplement (N), MCF-7 cell-conditioned medium (M), LNCap cell-conditioned medium (L), or HepG2 cell-conditioned medium (H). Data were scaled to bring all controls to equal values (25,000 cells, see ‘control’ line). Dashed lines show examples of linear dependence (n=k∗r +b) between cytotoxicity assay data *n* and *r*. All data in the plot were described by linear equations, and variations of the coefficients (*k* and *b*) were interdependent. (**b**) Coefficients of linear equations that describe the dependence of cytotoxicity on *r*. “*k*” and “*b*” values correspond to coefficients of linear approximations showed on plot a. Equation of linear approximation of the coordinates is shown on the plot. Adapted from [90,91].

**Figure 6 vaccines-07-00186-f006:**
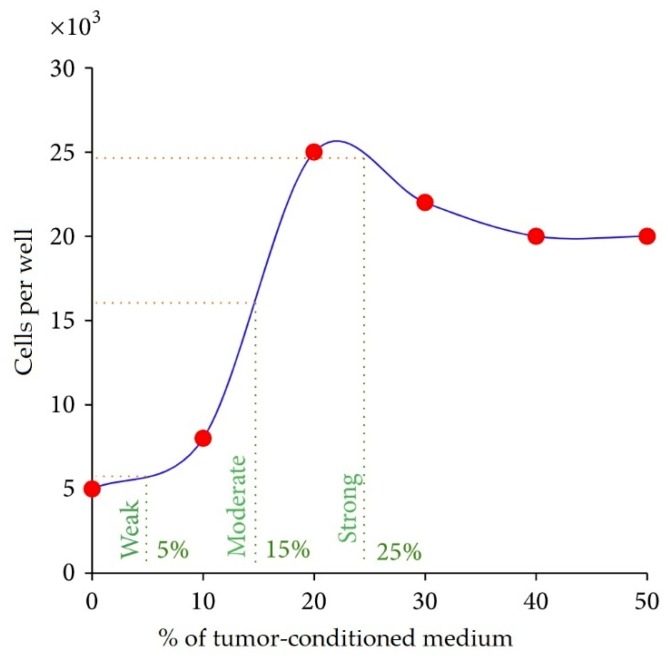
Dose determination of tumor-conditioned medium used to prepare HMECs with tumor-induced cell surface profiles. HMEC cultures were incubated with 0%, 10%, 20%, 30%, 40%, or 50% tumor-conditioned medium. After three days in culture, cells were counted (red points) in wells using trypan blue exclusion. Cell numbers were approximated using a curve used to determine the concentrations of the tumor-conditioned medium that elicited either weak (stimuli slightly higher than in the control), moderate (half of the maximum), or strong (maximum) stimulation of HMEC cultures (green lines). Green line labels indicate the percentage of the tumor-conditioned medium selected for CTA. Adapted from [92].

**Figure 7 vaccines-07-00186-f007:**
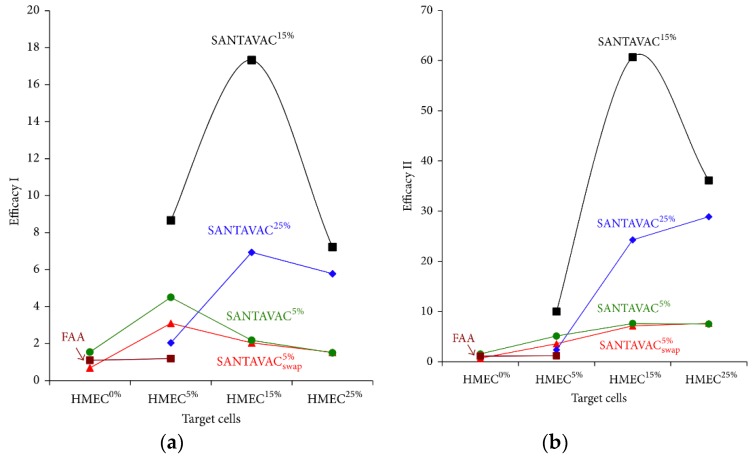
Efficacy of target cell killing by SANTAVAC in cytotoxicity assays (CTA). (**a**) Data are expressed as efficacy I of target cell killing by SANTAVAC. (**b**) Data are expressed as efficacy II of target cell killing by SANTAVAC. Target HMECs were incubated in the presence of effector CTLs at a 1:20 ratio. After three days, CTLs were removed, and target cell viability was determined. Efficacy I was calculated as a ratio of the number of non-stimulated cells in control (i.e., HMEC^0%^) to the number of tumor-stimulated cells in the experiment. Efficacy II was calculated as a ratio of the number of tumor-stimulated cells in control (i.e., HMEC^5%^, HMEC^15%^, or HMEC^25%^) to the number of tumor-stimulated cells in the experiment, that is, the percentage of the tumor-conditioned medium in control was the same as in the experiment. Efficacy I allows in vitro estimation of the SANTAVAC efficacy by demonstrating how many endothelial cells in the tumor vasculature will be destroyed before one endothelial cell in normal tissue is destroyed (used to predict vaccine safety). Efficacy II allows in vitro estimation of the SANTAVAC efficacy by demonstrating the degree of HMEC proliferation suppression in the tumor vasculature and is used to establish the degree by which the vaccine can arrest tumor growth (predicted vaccine therapeutic effect). For efficacy calculation, the data representing the mean value of three independent measurements was used. “FAA” indicates the data related to the control (■) in CTA where fibroblast-associated antigens were used to simulate CTL. “swap” (▲) indicates CTA data where primary cell cultures used to generate SANTAVAC and primary cell cultures which were used as target cells were swapped (it was done to demonstrate the reproducibility of the CTA results at defined percentages of the tumor-conditioned medium used to stimulate HMEC). Percentage values indicated in the superscript and label shape correspond to the percentage (●—5%, ■—15%, ♦—25%) of the tumor-conditioned medium used to stimulate target HMEC or HMEC used to generate SANTAVAC for targeting immune response in CTA. Adapted from [92].

**Figure 8 vaccines-07-00186-f008:**
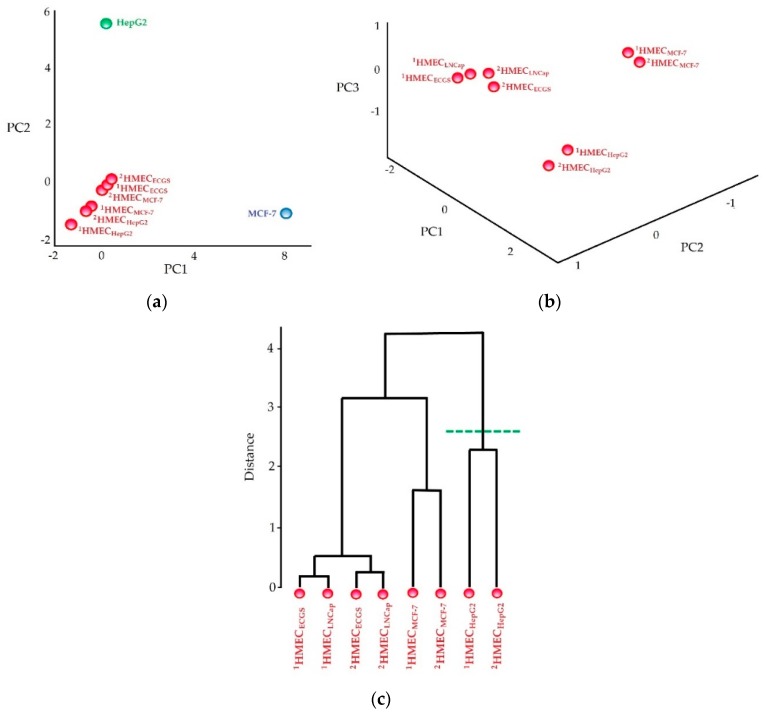
Authentication of antiangiogenic SANTAVAC composition. (**a**) PCA of SANTAVAC codes obtained for HMECs and cancer cells, (**b**) PCA of SANTAVAC codes obtained only for HMECs, (**c**) dendrogram depicting the distances measured between points showed on plot b. “^1^HMEC_ECGS_” and “^2^HMEC_ECGS_”: HMECs stimulated by endothelial cell growth supplement (i.e., by signal from normal cells), “^1^HMEC_MCF-7_” and “^2^HMEC_MCF-7_”: HMECs stimulated by MCF-7 cancer cells, “^1^HMEC_LNCap_” and “^2^HMEC_LNCap_”: HMECs stimulated by LNCap cancer cells, “^1^HMEC_HepG2_” and “^2^HMEC_HepG2_”: HMECs stimulated by HepG2 cells, “MCF-7”: MCF-7 cells, “HepG2”: HepG2 cells. Green line indicates the cluster considered to be antiangiogenic SANTAVAC, which is qualified for vaccination. Adapted from [90].

**Table 1 vaccines-07-00186-t001:** SANTAVAC specifications.

Property	Value
Appearance (turbidity)	Clear
Appearance (color)	Clear
Appearance (form)	Solution
pH	7.2–7.6
Osmolality	260–300 mOs/kg
Salt composition	Hank’s balanced salt solution
Endotoxin level	<1.0 EU/mL
Cytotoxicity overlay	Non-toxic
Polypeptides concentration	2.5–3 mg/L
Trypsin autolysis products	traces
Activity	0.8–0.9 U/mL ^1^

^1^ One unit corresponds to antigens obtained from one million cells.

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
