# Peer review of "SANTAVAC^TM^: Summary of Research and Development"

_vaccines, 2019, doi:10.3390/vaccines7040186_

Round 1
Reviewer 1 Report
The manuscript from Lokhov et al. describes in detail the developmet of SANTAVAC, a formulation of antigens aimed at obtaining cancer vaccines. The literature regarding SANTAVAC is rich and complete, the technology is well described and all the in vitro evidences of the potential efficacy of the use of SANTAVAC is reported. Nevertheless the review sounds more as a description of a product rather than a global view of the context in which SANTAVAC could have a clinical impact. Thus, the authors may:
better delineate the status quo of cancer vaccines, indicating the potential improvement that the use of SANTAVAC can give, in relation to the current anticancer vaccinations; better explain the implications of targeting endothelial cells and tumor angiogenesis. The authors should insert more recent citations on tumor angiogenesis and antiangiogenic therapies, discussing the controversial efficacy of such therapeutic strategies.Author Response
Dear Reviewer,
Thank you for your suggestions regarding our manuscript: “SANTAVACTM: Summary of Research & Development". We appreciate your feedback and have addressed each of your comments below.
Remark: “the authors may better delineate the status quo of cancer vaccines, indicating the potential improvement that the use of SANTAVAC can give, in relation to the current anticancer vaccinations”
Response: The revised introduction begins with two paragraphs which briefly describe the status quo of cancer vaccines. The following paragraphs indicate what SANTAVAC can offer in relation to the current options.
Remark: “The authors should insert more recent citations on tumor angiogenesis and antiangiogenic therapies, discussing the controversial efficacy of such therapeutic strategies”.
Response: Section 6.1, “Background, Rationale and Significance”, was added and in this section the requested citations are provided. The controversial efficacy of such therapeutic strategies is also mentioned in this section.
The authors are grateful to the Reviewer for a very high appreciation of the work.
Sincerely,
Dr. Lokhov
Reviewer 2 Report
This interesting review reports on SANTAVAC vaccine, which is a vaccine composed of cell surface proteins from endothelial cells. This is indeed a highly innovative and novel approach, and preclinical results are encouraging. The linguistic style is very good, which makes the manuscript easy to read and understand. This review will be of high interest to many readers. Therefore I support publication in "vaccines".
Prior to publication, following issues must be adressed however:
Please change the the wording "...achieved efficacy equal to 17 and 60 in relation to prediction..." in the abstract and on page 17. It is not clear what you mean here. Please describe the following steps you have planned in the development of this vaccine: preclinical experiments in a mouse model? An early clinical study? Please make clear that although the results you have presented are promising, that this vaccine should not be used outside clinical studies unless unequivocal data from clinical studies is available.Author Response
Dear Reviewer,
Thank you for your suggestions regarding our manuscript: “SANTAVACTM: Summary of Research & Development". We appreciate your feedback and have addressed each of your comments below.
Remark: “Please change the wording "...achieved efficacy equal to 17 and 60 in relation to prediction..." in the abstract and on page 17. It is not clear what you mean here”.
Response: Both sentences were rephrased.
Remark: “Please describe the following steps you have planned in the development of this vaccine: preclinical experiments in a mouse model? An early clinical study?”
Response: This information was added at the end of section 9, “Development Plans”.
Remark: “Please make clear that although the results you have presented are promising, that this vaccine should not be used outside clinical studies unless unequivocal data from clinical studies is available”.
Response: This point is now included in section 9, “Development Plans”.
The authors are grateful to the Reviewer for a very high appreciation of the work.
Sincerely,
Dr. Lokhov
Reviewer 3 Report
The authors describe the product development SANTAVAC, an antigen composition developed via proteomics and cell culture technology that is intended for the development of cancer vaccines against various solid tumors. They propose that its mechanism of action is based on the heterogeneity of endothelial cells, the polypeptides of which are similar to the surface antigens of tumor-vessel cells, allowing targeted destruction by vaccination.
Although the authors did a lot of work on this antigenic composition, I am not convinced how this is mechanistically working, taking into consideration antigen presentation via dendritic or tumor cells for CTL activity and the heterogeneity of MHC molecules. It is not clear whether this antigenic composition is adequately presented to T cells. It makes more sense to use MHC-associated peptide proteomics to identify immunogenic epitopes than whole antigens.
Author Response
Dear Reviewer,
Thank you for your suggestions regarding our manuscript: “SANTAVACTM: Summary of Research & Development". We appreciate your feedback and have addressed each of your comments below.
Remark: “Although the authors did a lot of work on this antigenic composition, I am not convinced how this is mechanistically working, taking into consideration antigen presentation via dendritic or tumor cells for CTL activity and the heterogeneity of MHC molecules. It is not clear whether this antigenic composition is adequately presented to T cells”.
Response: Thank you for this remark. The author believes that at the current stage of vaccine development, the data provided in the article are sufficient. The killing rate of target cells in cytotoxicity assays (presented in all of our SANTAVAC-related research papers) is an integrative parameter which already takes into account antigen presentation by MHC. Regarding in vivo tests of final products, they will proceed according to FDA requirements, which currently include the estimation of T-cell infiltration into the tumor and actual activation of T cells. We are ready to proceed with the MHC- and T cell-related experiments once they are requested by the FDA.
Remark: “It makes more sense to use MHC-associated peptide proteomics to identify immunogenic epitopes than whole antigens.”
Response: Thank you for this remark. There are many methods for drug development, each with their own advantages and shortcomings. Authors agree that MHC-associated peptide proteomics is a very promising approach, and in the future it will likely be used for the preparation of immunogenic epitopes from SANTAVAC products. But the authors believe that there are no compelling reasons to counterpose this approach to the proteomics currently used in SANTAVAC development. It is unclear whether MHC-associated peptides would be free of the shortcomings associated with antiangiogenic vaccines based on separate antigens, or whether these peptides would provide confirmation of their efficiency with mathematical precision. The nature of SANTAVAC, with its diversity of surface antigens tempering the fluctuations due to individual peptides, lends itself well to the methods employed here.
Sincerely,
Dr. Lokhov
Reviewer 4 Report
The manuscript entitled “ SANTAVACTM: Summary of Research & 2 Development” well describes the current state of scientific knowledge regarding innovation, background, description of product development, and summary of nonclinical studies with SANTAVAC to date are presented in this review. SANTAVAC is an antigen composition developed via proteomics and cell culture technology that is intended for the development of cancer vaccines against various solid tumors. Its mechanism of action is based on the heterogeneity of endothelial cells, the polypeptides of which are similar to the surface antigens of tumor-vessel cells, allowing targeted destruction by vaccination. While research and development work with SANTAVAC is ongoing, the existing data provide strong evidence that allogeneic SANTAVAC is an ideal candidate for the development of cancer vaccines with significant efficacy and safety. The SANTAVAC formulations described here achieved efficacy equal to 17 and 60 in relation to prediction of vaccine safety and capacity to arrest tumor growth, respectively. It provides insights into the clinical implementation.
Author Response
The authors are grateful to the Reviewer for a very high rating of the article.
No answers have been provided since the Reviewer has no comments and suggestions on the text of the article.
Sincerely,
Dr. Lokhov
Round 2
Reviewer 3 Report
I propose acceptance of the manuscript